

# Properties of heavy mesons at finite temperature

Glòria Montaña[1★], Àngels Ramos[1] and Laura Tolós[2,3,4,5]

1 Departament de Física Quàntica i Astrofísica and Institut de Ciències del Cosmos (ICCUB),
Facultat de Física, Universitat de Barcelona, Martí i Franquès 1, 08028 Barcelona, Spain
2 Institut für Theoretische Physik, Goethe Universität Frankfurt,
Max von Laue Strasse 1, 60438 Frankfurt, Germany
3 Frankfurt Institute for Advanced Studies, Goethe Universität Frankfurt,
Ruth-Moufang-Str. 1, 60438 Frankfurt am Main, Germany
4 Institute of Space Sciences (ICE, CSIC), Campus UAB,
Carrer de Can Magrans, 08193, Barcelona, Spain
5 Institut d'Estudis Espacials de Catalunya (IEEC), 08034 Barcelona, Spain

★ gmontana@fqa.ub.edu

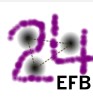

*Proceedings for the 24th edition of European Few Body Conference,
Surrey, UK, 2-6 September 2019*

## Abstract

We study the properties of heavy mesons using a unitarized approach in a hot pionic medium, based on an effective hadronic theory. The interaction between the heavy mesons and pseudoscalar Goldstone bosons is described by a chiral Lagrangian at next-to-leading order in the chiral expansion and leading order in the heavy-quark mass expansion so as to satisfy heavy-quark spin symmetry. The meson-meson scattering problem in coupled channels with finite-temperature corrections is solved in a self-consistent manner. Our results show that the masses of the ground-state charmed mesons $D(0^-)$ and $D_s(1^-)$ decrease in a pionic environment at $T \neq 0$ and they acquire a substantial width. As a consequence, the behaviour of excited mesonic states (i.e. $D_{s0}^*(2317)^{\pm}$ and $D_0^*(2300)^{0,\pm}$), generated dynamically in our heavy-light molecular model, is also modified at $T \neq 0$. The aim is to test our results against Lattice QCD calculations in the future.

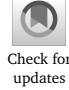

## 1 Introduction

Relativistic heavy-ion collisions offer a unique scenario to study the production of heavy mesons and multiquark states in general in extreme conditions of temperature and density. Heavy mesons, i.e., charmed and bottom mesons, are of particular interest in this respect since heavy flavour is mainly produced at the early stages of the heavy-ion collisions and hence they are

usually considered the ideal probes of the quark-gluon plasma (QGP). After it is produced, the heavy flavour interacts with the hot dense matter, first with the QGP and with the hot hadronic medium after hadronization. Therefore, there is a need for a better theoretical understanding of the properties of heavy mesons at temperatures and densities far from the nuclear regime.

In this contribution, we focus on the study of the QCD phase diagram in the high-temperature and low-density regime, which corresponds to matter generated in heavy-ion collisions at the Relativistic-Heavy-Ion-Collider (RHIC) at BNL and at the Large-Hadron-Collider (LHC) at CERN. For this reason we consider mesonic matter at finite temperature, which can be well-approximated to be mainly pionic at temperatures below the critical temperature for the transition from the deconfined QGP to the hadron gas, $T_c$.

We present results of the modification of the properties of the $D$-mesons ($D^{(*)0}$, $D^{(*)+}$, $D_s^{(*)+}$) when interacting with the surrounding pions in such a hot environment. We also show results for the excited mesonic states dynamically generated in a heavy-light molecular model. In particular, we study the $D_{s0}^*(2317)^{\pm}$ and the $D_0^*(2300)^{0,\pm}$, that are the lightest strange and non-strange excited mesons, respectively, and which have attracted much attention within the molecular models as their masses differ from the quark model expectations.

## 2 Formalism

### 2.1 Interaction of heavy mesons with light mesons

The Lagrangian density that describes the interaction between $D^{(*)}$- and $D_s^{(*)}$-mesons with spin-parity $J^P = 0^-(1^-)$ and pseudoscalar Goldstone bosons ($\pi$, $K$, $\bar{K}$ and $\eta$) to next-to-leading order (NLO) in the chiral expansion and keeping the leading order (LO) in the heavy-quark mass expansion is given by:

$$\mathcal{L} = \mathcal{L}_{\text{LO}} + \mathcal{L}_{\text{NLO}}. \tag{1}$$

The LO contribution contains the kinetic and the mass terms of the heavy mesons as well as interaction terms [1–4]:

$$\mathcal{L}_{\text{LO}} = \langle \nabla^\mu D \nabla_\mu D^\dagger \rangle - m_D^2 \langle D D^\dagger \rangle - \langle \nabla^\mu D^{*\nu} \nabla_\mu D_\nu^{*\dagger} \rangle + m_D^2 \langle D^{*\nu} D_\nu^{*\dagger} \rangle$$
$$+ ig \langle D^{*\mu} u_\mu D^\dagger - D u^\mu D_\mu^{*\dagger} \rangle + \frac{g}{2m_D} \langle D_\mu^* u_\alpha \nabla_\beta D_\nu^{*\dagger} - \nabla_\beta D_\mu^* u_\alpha D_\nu^{*\dagger} \rangle \epsilon^{\mu\nu\alpha\beta}, \tag{2}$$

where the brackets, $\langle \cdots \rangle$, denote the trace in flavour space, $m_D$ is the mass in the chiral limit of the heavy mesons and the heavy-light pseudoscalar-vector coupling constant $g$ is the same for the two interaction terms considering heavy-quark spin symmetry (HQSS). The $D$ and $D_\mu^*$ are the $J^P = 0^-, 1^-$ SU(3) antitriplets $\left(D = \begin{pmatrix} D^0 & D^+ & D_s^+ \end{pmatrix}, D_\mu^* = \begin{pmatrix} D^{*0} & D^{*+} & D_s^{*+} \end{pmatrix}_\mu\right)$ and $\nabla_\mu D^{(*)} = \partial_\mu D^{(*)} - D^{(*)} \Gamma^\mu$ is their covariant derivative. The vector and axial-vector currents are $\Gamma_\mu = \frac{1}{2}(u^\dagger \partial_\mu u + u \partial_\mu u^\dagger)$ and $u_\mu = i(u^\dagger \partial_\mu u - u \partial_\mu u^\dagger)$, respectively, with $u = \sqrt{U} = \exp(\frac{i\Phi}{\sqrt{2}f_\pi})$ and $\Phi$ the $(3 \times 3)$-matrix encoding the octet of Goldstone boson fields,

$$\Phi = \begin{pmatrix} \frac{1}{\sqrt{2}}\pi^0 + \frac{1}{\sqrt{6}}\eta & \pi^+ & K^+ \\ \pi^- & -\frac{1}{\sqrt{2}}\pi^0 + \frac{1}{\sqrt{6}}\eta & K^0 \\ K^- & \bar{K}^0 & -\sqrt{\frac{2}{3}}\eta \end{pmatrix}, \tag{3}$$

with $f_\pi = 92.4\,\text{MeV}$ the pseudoscalar decay constant in the chiral limit.

The NLO chiral Lagrangian term reads [5–8]

$$
\begin{aligned}
\mathcal{L}_{\text{NLO}} =& -h_0\langle DD^\dagger\rangle\langle\chi_+\rangle + h_1\langle D\chi_+ D^\dagger\rangle + h_2\langle DD^\dagger\rangle\langle u^\mu u_\mu\rangle \\
&+ h_3\langle Du^\mu u_\mu D^\dagger\rangle + h_4\langle\nabla_\mu D\nabla_\nu D^\dagger\rangle\langle u^\mu u^\nu\rangle + h_5\langle\nabla_\mu D\{u^\mu,u^\nu\}\nabla_\nu D^\dagger\rangle \\
&+ \tilde{h}_0\langle D^{*\mu}D_\mu^{*\dagger}\rangle\langle\chi_+\rangle - \tilde{h}_1\langle D^{*\mu}\chi_+ D_\mu^{*\dagger}\rangle - \tilde{h}_2\langle D^{*\mu}D_\mu^{*\dagger}\rangle\langle u^\nu u_\nu\rangle \\
&- \tilde{h}_3\langle D^{*\mu}u^\nu u_\nu D_\mu^{*\dagger}\rangle - \tilde{h}_4\langle\nabla_\mu D^{*\alpha}\nabla_\nu D_\alpha^{*\dagger}\rangle\langle u^\mu u^\nu\rangle - \tilde{h}_5\langle\nabla_\mu D^{*\alpha}\{u^\mu,u^\nu\}\nabla_\nu D_\alpha^{*\dagger}\rangle, \quad (4)
\end{aligned}
$$

where $\chi_+ = u^\dagger\chi u^\dagger + u\chi u$ with the quark mass matrix $\chi = \text{diag}(m_\pi^2, m_\pi^2, 2m_K^2 - m_\pi^2)$.

At LO in the heavy-quark expansion the equality $\tilde{h}_i = h_i$ ($i = 0,1,...,5$) holds for the low-energy constants (LECs), the value of which can be fitted to LQCD data. In this case, the tree-level scattering amplitude of a $D^{(*)}$- or $D_s^{(*)}$-meson scattered with a light meson reads

$$
\begin{aligned}
V^{jk}(s,t,u) =& \frac{1}{f_\pi^2}\Big[\frac{C_{\text{LO}}^{jk}}{4}(s-u) - 4C_0^{jk}h_0 + 2C_1^{jk}h_1 \\
&- 2C_{24}^{jk}\Big(2h_2(p_2\cdot p_4) + h_4\big((p_1\cdot p_2)(p_3\cdot p_4) + (p_1\cdot p_4)(p_2\cdot p_3)\big)\Big) \\
&+ 2C_{35}^{jk}\Big(h_3(p_2\cdot p_4) + h_5\big((p_1\cdot p_2)(p_3\cdot p_4) + (p_1\cdot p_4)(p_2\cdot p_3)\big)\Big)\Big],
\end{aligned} \quad (5)
$$

where $p_1$ and $p_2$ ($p_3$ and $p_4$) are the momenta of the incoming (outgoing) mesons and $C_{\text{LO},0,1,24,35}$ are the isospin coefficients (see Table II in [5]). The $j,k$ indices denote channels in the sector with charm $C$, strangeness $S$ and isospin $I$ in the isospin basis, as isospin violation is not considered.

## 2.2 Unitarized amplitudes at $T = 0$

The interaction above is unitarized through the Bethe-Salpeter (BS) approach in coupled channels describing the two-body scattering in a basis with several channels, which in its matrix form is written as $T = V + VGT$. This equation has a purely algebraic solution for the on-shell resummed amplitude:

$$
T(s) = V(s)[1 - V(s)G(s)]^{-1}, \quad (6)
$$

where $V(s)$ is the matrix containing the interaction potentials of Eq. (5) and $G(s)$ is the diagonal matrix constructed from the meson-meson loop functions, with the characteristic unitarity cut above threshold and regularized with a cutoff.

The analytical continuation of Eq. (6) to the complex-energy plane allows to identify quasibound, resonant and virtual states from poles in different Riemann sheets (RS) of the $T$-matrix. In addition to the pole position given by the mass, $M_R = \text{Re}\sqrt{s_R}$, and the width, $\Gamma_R/2 = \text{Im}\sqrt{s_R}$, one can also obtain the coupling, $g_i$, of the pole to the channel $i$ from the residue around the pole position, which is associated to the strength of that channel in the generation of the resonance, and the compositeness, $X_i = |g_i|^2|\partial G_i/\partial s|_{s=s_R}$, that can be interpreted as the importance of the two-meson channel $i$ component in the dynamically generated state.

## 2.3 Finite temperature

The main novelty of the work presented in this contribution is the extension of the model above so as to include finite temperature corrections. We use the method described in [9,10] up to some technicalities. On the one hand, in order to take into account the effect of a pionic bath at finite temperature on the properties of the ground-state charmed mesons, we use the imaginary time formalism (ITF). It essentially consists in replacing the real energy of the propagator by discrete imaginary frequencies, $q^0 \to \omega_n = i2\pi nT$ (for bosons), commonly

referred to as Matsubara frequencies, and the corresponding intermediate energy integrals by sums over discrete values, namely,

$$\int \frac{d^4q}{(2\pi^4)} \rightarrow \frac{i}{\beta} \sum_n \int \frac{d^3q}{(2\pi)^3}. \tag{7}$$

On the other hand, the self-energy of the heavy meson (Fig. 1a), obtained from closing the pion line in the $T$-matrix element corresponding to $D_{(s)}\pi \rightarrow D_{(s)}\pi$ scattering (Fig. 1c), is employed to dress its propagator (Fig. 1b).

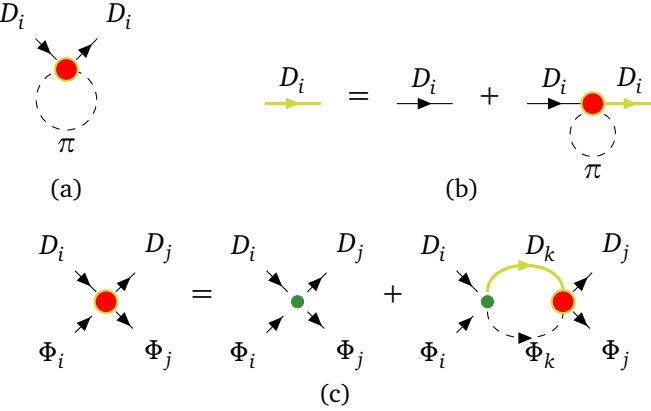

Figure 1: Diagramatic representation of (a) the self-energy of the $D_{(s)}$-meson, (b) the dressed $D_{(s)}$-meson propagator and (c) the BS equation at $T \neq 0$ with dressed propagators in the loop.

Thus, upon summation over the Matsubara frequencies and extrapolation to the real axis, the most general expression that we can write for the two-meson loop function at finite temperature is:

$$G_{D_i \Phi_i}(E, \vec{p}; T) = \int \frac{d^3q}{(2\pi)^3} \int d\omega \int d\omega' \frac{S_{D_i}(\omega, \vec{q}; T) S_{\Phi_i}(\omega', \vec{p}-\vec{q}; T)}{E - \omega - \omega' + i\varepsilon} [1 + f(\omega, T) + f(\omega', T)], \tag{8}$$

where $f(\omega, T)$ and $f(\omega', T)$ are the Bose-Einstein distributions and $S_{D_i}$ and $S_{\Phi_i}$ the spectral functions, defined below. As in the zero temperature case, we regularize the loop function above using a momentum cutoff. This expression can be simplified if we do not consider the modification of the light meson by its interaction with the pions in the bath and hence its spectral function, $S_{\Phi_i}$, can be replaced with a delta function. The results of the direct numerical integration of the resulting expression are displayed in Figure 4.

The ITF leads to a factor containing a combination of meson Bose distribution functions, $f(\omega, T) = (e^{\omega/T} - 1)^{-1}$, at temperature $T$ in Eq. (8). To give a physical interpretation, it is useful to change the limits of integration to $[0, \infty)$ and split the integral into four terms, as well as to rewrite the sums of Bose functions. After some analytical work one can show that the integrand reads (with the notation $f_{D_i} := f(\omega, T)$ and $f_{\Phi_i} := f(\omega', T)$ for simplicity)

$$\frac{[1 + f_{D_i}][1 + f_{\Phi_i}] - f_{D_i} f_{\Phi_i}}{E - \omega - \omega' + i\varepsilon} + \frac{f_{D_i} f_{\Phi_i} - [1 + f_{D_i}][1 + f_{\Phi_i}]}{E + \omega + \omega' + i\varepsilon}$$
$$+ \frac{f_{D_i}[1 + f_{\Phi_i}] - f_{\Phi_i}[1 + f_{D_i}]}{E + \omega - \omega' + i\varepsilon} + \frac{f_{\Phi_i}[1 + f_{D_i}] - f_{D_i}[1 + f_{\Phi_i}]}{E - \omega + \omega' + i\varepsilon}, \tag{9}$$

up to a factor containing the spectral functions. For example, the first term may be interpreted as the production of a heavy-light meson pair, occurring with a statistical weight factor $(1 + f_{D_i})(1 + f_{\Phi_i})$, minus the absorption of a heavy-light meson pair, which is possible at finite

temperature, with a statistical weight factor $f_{D_i} f_{\Phi_i}$. Similarly, the other terms can be related to production and absorption processes of particles and antiparticles by the bath [11,12]. We note that at $T = 0$ the Bose distribution function vanishes and only the terms corresponding to the production of a meson-meson pair and an antimeson-antimeson pair survive.

The location of branch cuts can also be read out from Eq. (9). In addition to the standard unitarity cut above the channel threshold, $E \geq (m_{D_i} + m_{\Phi_i})$, which is the standard cut at $T = 0$, because of the additional processes that are allowed at finite temperature a new cut develops for $E \leq |m_{D_i} - m_{\Phi_i}|$. This is known as the Landau cut and has no counterpart in the vacuum theory. In Appendix A we show results for the loop functions of the various heavy-light channels where one can appreciate these features at different temperatures.

The unitarized $T$-matrix resulting from solving the BS equation with loops including finite temperature corrections but undressed heavy mesons is employed to obtain the first iteration of the self-energy of the heavy mesons in a pionic bath:

$$\Pi_{D_i}(E, \vec{p}; T) = \int \frac{d^3 q}{(2\pi)^3} \int d\Omega \, \frac{E}{\omega_\pi} \frac{f(\Omega, T) - f(\omega_\pi, T)}{E^2 - (\omega_\pi - \Omega)^2 + i\varepsilon} \left( -\frac{1}{\pi} \right) \mathrm{Im}\, T_{D_i \pi}(\Omega, \vec{p} + \vec{q}; T), \qquad (10)$$

where again there is a combination of Bose factors coming from the ITF. The spectral function is then readily obtained from the imaginary part of the dressed propagator (Fig. 1b) as

$$S_{D_i}(\omega, \vec{q}; T) = -\frac{1}{\pi} \mathrm{Im}\, \mathcal{D}_{D_i}(\omega, \vec{q}; T) = -\frac{1}{\pi} \mathrm{Im} \left( \frac{1}{\omega^2 - \vec{q}^2 - m_{D_i}^2 - \Pi_{D_i}(\omega, \vec{q}; T)} \right), \qquad (11)$$

and used in the loop function of Eq. (8). The new unitarized amplitude calculated using dressed loops (Fig. 1c) modifies, in turn, the self-energy (Fig. 1a) and therefore this procedure needs to be iterated several times to ensure self-consistent results.

## 3 Results

We study here the heavy-light meson scattering in the sectors with charm, strangeness and isospin $(C, S, I) = (1, 0, 1/2)$ and $(1, 1, 0)$, where there are strong indications that the excited $D_0^*(2300)^{0,\pm}$ and $D_{s0}^*(2317)^\pm$ are dynamically generated in molecular models. We take the LECs of the NLO Lagrangian from Fit-2B in Ref. [8], but not the subtraction constants of the unitarization procedure that they also fit to lattice data, as we find that they might correspond to small and unrealistic values of the cutoff for certain channels, much smaller than the value of $\Lambda = 800$ MeV that we take. Also loop functions in dimensional regularization have positive real parts at low energies far from threshold that might generate unphysical poles in the $T$-matrix and make the numerical integration of Eq. (10) more complicated. We have checked that the obtained scattering lengths are also in excellent agreement with the lattice data.

### 3.1 Poles at $T = 0$

In Table 1, we give the pole positions and their couplings and compositeness for the coupled channels $D\pi(2005.3)$, $D\eta(2415.1)$ and $D_s\bar{K}(2464.0)$ (the threshold energies in parenthesis) in the $(1, 0, 1/2)$ sector and $DK(2364.9)$ and $D_s\eta(2516.2)$ in the $(1, 1, 0)$ sector at zero temperature. Similarly to previous works [7,8], we find two poles in the $(1, 0, 1/2)$ sector that can be associated to the $D_0^*(2300)^{0,\pm}$. The lower pole, located just above the $D\pi$ threshold, is a resonance coupling mostly to $D\pi$ and qualifies as a $D\pi$ molecular state. The status of the higher pole is a bit more complicated. We find it above the $D_s\bar{K}$ threshold as a virtual state in the $(-,-,+)$ RS, but for some values of the parameters of the model [7,8] it appears as a resonant pole between the $D\eta$ and $D_s\bar{K}$ thresholds, strongly coupling to the $D_s\bar{K}$ channel in

both cases. In the $(1,1,0)$ sector we find a bound state for the $D_{s0}^*(2317)^\pm$, with a large $DK$ component.

Table 1: Poles and the corresponding couplings and compositeness for the coupled channels in the sectors with the quantum numbers of the $D_0^*(2300)^{0,\pm}$ (upper) and the $D_{s0}^*(2317)^\pm$ (lower).

| $(C,S,I)$ | RS | $M_R$ (MeV) | $\Gamma_R/2$ (MeV) | $|g_i|$ (GeV) | $X_i$ |
|---|---|---|---|---|---|
| $(1,0,1/2)$ | $(-,+,+)$ | 2081.9 | 86.0 | $|g_{D\pi}| = 8.9$ | $X_{D\pi} = 0.29 - i\,0.27$ |
| | | | | $|g_{D\eta}| = 0.4$ | $X_{D\eta} = 0.00 + i\,0.00$ |
| | | | | $|g_{D_s\bar{K}}| = 5.4$ | $X_{D_s\bar{K}} = 0.01 + i\,0.05$ |
| | $(-,-,+)$ | 2521.2 | 121.7 | $|g_{D\pi}| = 6.4$ | $X_{D\pi} = 0.02 + i\,0.09$ |
| | | | | $|g_{D\eta}| = 8.4$ | $X_{D\eta} = 0.15 - i\,0.27$ |
| | | | | $|g_{D_s\bar{K}}| = 14.0$ | $X_{D_s\bar{K}} = 0.43 + i\,0.49$ |
| $(1,1,0)$ | $(+,+)$ | 2252.5 | 0.0 | $|g_{DK}| = 13.3$ | $X_{DK} = 0.66 + i\,0.00$ |
| | | | | $|g_{D_s\eta}| = 9.2$ | $X_{D_s\eta} = 0.17 + i\,0.00$ |

## 3.2 Spectral functions of ground-state heavy mesons at $T \neq 0$

From the procedure described in Section 2.3 we obtain the modification of the ground-state properties of $D$- and $D_s$-mesons in a hot pionic bath as a function of the temperature. In Fig. 2 we display their zero-momentum spectral functions (see Eqs. (10) and (11)) for $T = 0, 100, 150\,\text{MeV}$. We show the experimental values of the masses in black (T=0). The main effects of the temperature are a shift of the peak, which can be associated to the mass at that temperature, towards lower values and a significant broadening with increasing temperatures.

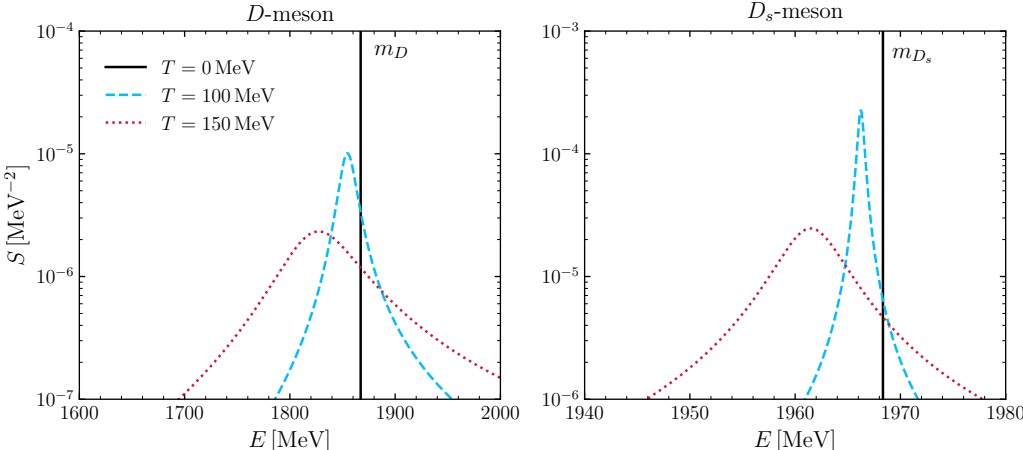

Figure 2: Spectral functions of the $D$- and $D_s$-mesons as a function of the energy, at zero momentum and at $T = 0, 100, 150\,\text{MeV}$.

The displacement of the peak is related to the non-zero real part of the self-energy at finite temperature, regularized by dropping the T=0 contribution, i.e. the terms without bose factor in the real part of Eq. (10) upon changing the limits of integration to $[0, \infty)$. We find a monotonic displacement with increasing temperatures that is about 2% and 0.5% of the mass for the $D$- and $D_s$-mesons, respectively, at $T = 150\,\text{MeV}$. Although small, we find this

result significant and contrasts with the negligible contribution of Re $\tilde{\Pi}_D$ compared to the mass found in [9]. In the case of the width, which is connected to the imaginary part, our results of $\Gamma_D(T = 150\,\text{MeV}) \sim 70\,\text{MeV}$ for the $D$-meson are comparable to those in [9], and in addition we obtain for the $D_s$-meson a width of $\Gamma_{D_s}(T = 150\,\text{MeV}) \sim 7\,\text{MeV}$.

We note that the lower modification with temperature of the $D_s$-meson with respect to the $D$-meson is basically related to its weaker interaction with pions, as the $D_s\pi$ interaction vanishes at LO (see [13] for more details).

An analogous study of the charmed vector mesons results in mass shifts of $\sim 2\%$ and $\sim 0.3\%$, and widths of $\sim 80\,\text{MeV}$ and $\sim 18\,\text{MeV}$ for the $D^*$- and $D_s^*$-mesons, respectively.

### 3.3  Unitarized amplitudes and excited states at $T \neq 0$

As shown in Section 3.1, the excited $D_0^*(2300)^{0,\pm}$ and the $D_{s0}^*(2317)^\pm$ are dynamically generated within our model at $T = 0$ as heavy-light molecules with $J^P = 0^+$. Therefore, the modification of the ground-state charmed mesons in a hot pionic bath has an immediate repercussion on the finite-temperature unitarized amplitudes.

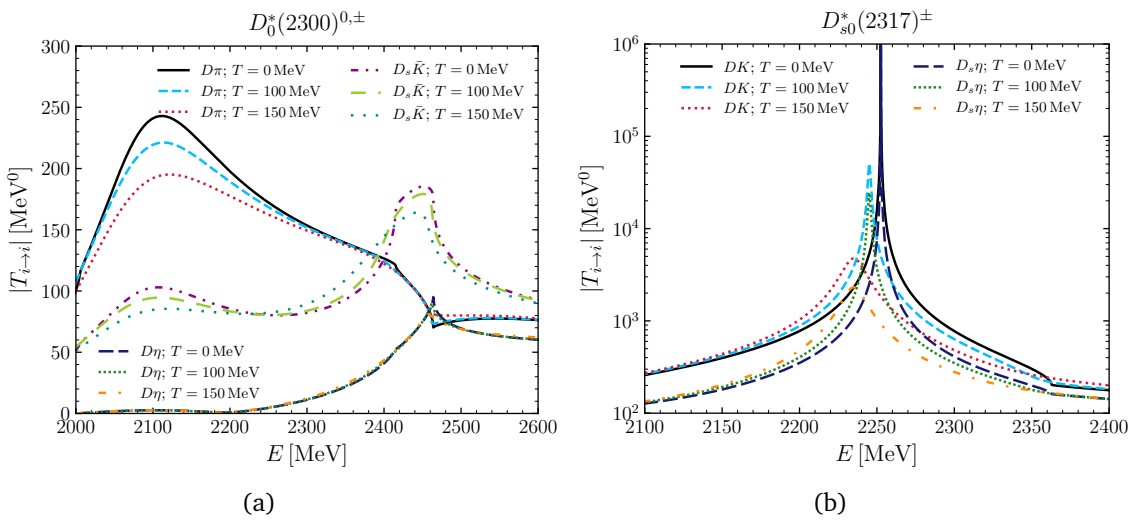

Figure 3: Absolute value of (a) the $D\pi \to D\pi$, $D\eta \to D\eta$ and $D_s\bar{K} \to D_s\bar{K}$ scattering amplitudes with $(C, S, I) = (1, 0, 1/2)$ and (b) the $DK \to DK$ and $D_s\eta \to D_s\eta$ ones with $(C, S, I) = (1, 1, 0)$ at $T = 0, 100, 150\,\text{MeV}$.

In Fig. 3 we show the absolute value of the diagonal $T$-matrix elements in the relevant sectors. In the $(1, 0, 1/2)$ sector we can identify the two resonances of the two-pole structure of the $D_0^*(2300)^{0,\pm}$ in the $D\pi \to D\pi$ and $D_s\bar{K} \to D_s\bar{K}$ amplitudes, respectively. For increasing temperatures the structures get diluted, as their widths increase in a pionic medium at finite temperature with respect to the vacuum.

Similarly, the delta function in the $T = 0$ amplitudes in the $(1, 1, 0)$ sector corresponding to a bound state for the $D_{s0}^*(2317)^\pm$ develops a finite width at $T \neq 0$. It moves up to $\sim 1\%$ towards lower energies and acquires a width of $\sim 15\,\text{MeV}$ at $T = 150\,\text{MeV}$.

## 4  Conclusions and Outlook

We have used a self-consistent formalism to study the effect of finite temperature on the scattering of charmed mesons off light mesons. In essence, our results show that the masses of the

pseudoscalar and vector ground-state charmed mesons decrease with increasing temperature while developing a substantial width in a hot pionic medium.

The modification of the $D^{(*)}$- and $D_s^{(*)}$-mesons has, in turn, consequences for the properties at finite temperature of resonances that are dynamically generated within our model as heavy-light molecules. We have indeed shown that the $D_0^*(2300)^{0,\pm}$, having basically a $D\pi$ and $D_s\bar{K}$ two-pole structure, and $D_{s0}^*(2317)^{\pm}$, which is mainly a $DK$ bound state at $T = 0$, get diluted as the temperature of the pion bath is increased. We will further discuss these results in a future paper [13]. It is interesting to note that the dilution of these excited states results from the relatively strong interaction of the charmed mesons with pions. Therefore these results are tied to the molecular description of our model for these excited states that would not be attained within quark models. These findings have to be considered when calculating heavy-ion collision observables.

In the near future we also expect to test our results against calculations from Lattice QCD. Furthermore, we also aim to study mesons with hidden-charm, as the $X(3872)$, as well as to extend our model to the bottom sector. We also plan to calculate transport properties at finite temperature, relevant for understanding the matter produced in heavy-ion collision experiments at the LHC or RHIC.

# Acknowledgements

G.M. and A.R. acknowledge support from the Spanish Ministerio de Economia y Competitividad (MINECO) under the project MDM-2014-0369 of ICCUB (Unidad de Excelencia 'María de Maeztu'), and, with additional European FEDER funds, under the contract FIS2017-87534-P. G.M. also acknowledges support from the FPU17/04910 Doctoral Grant from MINECO. L.T. acknowledges support from the FPA2016-81114-P Grant from Ministerio de Ciencia, Innovación y Universidades, Heisenberg Programme of the Deutsche Forschungsgemeinschaft under the Project Nr. 383452331 and THOR COST Action CA15213.

# A  Loop functions

In this appendix we briefly discuss the modification of the loop functions of heavy-light channels with temperature: $D\pi$, $DK$, $D\eta$ (Fig. 4a), $D_s\pi$, $D_sK$, $D_s\eta$ (Fig. 4b). The first thing to note is the two branch cuts described from Eq. (9) in Section 2.3, seen as kinks in the real parts (upper plots) and openings of the imaginary parts (lower plots) of the loops. As explained in the text, the interaction with the hot medium is responsible for the Landau cut at $E \leq |m_{D_i} - m_{\Phi_i}|$ and also for temperature corrections for the standard unitarity cut above threshold, the magnitude of which increases with increasing temperatures. Moreover the shift of the mass and widening of the heavy-meson spectral functions used for dressing the mesons in the loops produce shifts and smoothenings of the cuts, respectively. For this reason, the sharp unitarity cut at threshold observed at $T = 0$ and the kink of the real part become less abrupt with temperature.

As regards the differences among channels, the reason why the Landau cut is less pronounced for channels with heavier light mesons ($K$ and $\eta$) than for channels with a $\pi$ is that, because of their larger mass, they are scarce in the mesonic medium at the temperatures considered. Finally, the corrections resulting from the dressing of the $D_s$ are moderate compared to those in the case of the $D$ because its interaction with the pion is weaker and hence its modification with temperature is smaller.



**Figure 4:** Loop function for channels with (a) a $D$- or (b) a $D_s$-meson and a light meson, the real part in the upper subpanels and the imaginary part in the lower subpanels, at $T = 0, 100, 150\,\text{MeV}$.

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
