# Peer review of "Properties of heavy mesons at finite temperature"

_SciPost Physics Proceedings, doi:SciPost Phys. Proc. 3, 038 (2020)_

## Round 1 · Referee Report · Anonymous · 2020-1-13

Report

The proceedings contribution "Properties of heavy mesons at finite temperature" by Montaña, Ramos and Tolos provides a very clear summary of recent progress of the theoretical description of heavy mesons in a hot pion environment at equilibrium and in finite-temperature conditions. I feel the write-up is very clear, with a transparent justification, an illustrative description of the theoretical techniques and a succinct description of relevant results (including an analysis of loops in the appendix). I feel this goes beyond expectations in a proceedings contribution and fully endorse the publication.

Requested changes

I do have however a list of minor requests for clarification that I hope the authors can incorporate at the proofing stage.

1) Eq. (5): I don't think the C_LO, C_24 and C_35 constants have been introduced before. Could the authors provide a brief description of their meaning?
2) Immediately before Eq. (8): "the most general expression" seems to indicate there is a freedom of choice in getting Eq. (8). Is this the case, or is the expression dictated by IFT? Along these lines, I feel a general readership would benefit from references on how these integrals are computed (eg direct numerical integration or using the techniques outlined in Ref [9]?).
3) Eq. (9): is there a missing \omega in the first term of the second line?

I recommend publication of this comprehensive and clear contribution for the European Few-Body Conference, on the basis that it provides a clear description of timely theoretical work in hadronic physics.

---

## Editorial Decision

published